# Calcium Antagonist-Induced Gingival Overgrowth: A Case Report and Literature Review

**DOI:** 10.3390/diagnostics15030320

**Published:** 2025-01-30

**Authors:** Stefano Speroni, Marco Giuffrè, Tommaso Tura, Qamar Ammar Salman Al Jawaheri, Luca Antonelli, Luca Coccoluto, Giulia Bortune, Francesco Sarnelli, Silvio Abati

**Affiliations:** 1Dental School, Vita-Salute San Raffaele University, IRCCS San Raffaele, 20132 Milan, Italy; m.giuffre3@studenti.unisr.it (M.G.); t.tura@studenti.unisr.it (T.T.); q.aljawaheri@studenti.unisr.it (Q.A.S.A.J.); l.antonelli1@studenti.unisr.it (L.A.); l.coccoluto@studenti.unisr.it (L.C.); g.bortune@studenti.unisr.it (G.B.); f.sarnelli@studenti.unisr.it (F.S.); abati.silvio@hsr.it (S.A.); 2Department of Dentistry, IRCCS San Raffaele Hospital and Dental School, Vita-Salute San Raffaele University, 20123 Milan, Italy

**Keywords:** calcium channel blockers, drug-induced gingival overgrowth (DIGO), amlodipine, gingival overgrowth, histopathology

## Abstract

**Background**: Drug-induced gingival enlargement is a commonly documented adverse effect in patients administered with calcium antagonist medications. Nifedipine is the medicine most frequently linked to instances of gingival enlargement; nevertheless, amlodipine, likewise a calcium antagonist, can elicit this adverse effect. This case report aims to detail a case of amlodipine-induced gingival hyperplasia, emphasizing the significance of a multidisciplinary approach and outlining its therapy across various surgical phases. **Methods**: A 48-year-old hypertensive patient using amlodipine therapy presents with aberrant gingival tissue growth in the upper arch. Intraoral examination reveals localized inflammation and tissue enlargement in the papillae areas of the upper arch gingiva, leading to partial covering of the dental crowns. The patient experienced painful sensations and episodes of spontaneous bleeding in the enlarged gingival tissue. Following an initial professional dental hygiene treatment, which included root planning in the upper quadrants, and in consultation with the referring cardiologist, it was determined to discontinue amlodipine and initiate a replacement therapy with olmesartan medoxomil. Fifteen days following the cessation of amlodipine, surgical excision of the thickened interdental gingival tissues in the anterior region was conducted to obtain biopsies for histological confirmation of the observed pathological condition. **Results**: Histopathological examination validated the diagnosis of drug-induced gingival enlargement, characterized by chorion fibrosis and significant lymphoplasmacytic infiltration. Specifically, parakeratotic and acanthotic characteristics were seen in the gingival epithelium. Adjacent to the inflammatory regions, fibrosis was noted, along with the presence of cytoid bodies, which are typically linked to pathological diseases driven by inflammatory processes. These histological characteristics were consistent with the diagnosis of drug-induced gingival enlargement. **Conclusions**: A multidisciplinary approach involving the treating physician, dentist, and hygienist, incorporating drug replacement and targeted oral hygiene sessions, is crucial for the management and resolution of calcium channel blocker-induced gingival enlargement.

## 1. Introduction

Gingival overgrowth is characterized by a generalized increase in volume, often exhibiting minimal vascularity, with a smooth or nodular appearance of the superficial mucosa, which may occasionally result in nearly full coverage of dental tissues [1]. The presentation may range from subtle to pronounced proliferation of the gingival tissues of the papillae and/or marginal gingiva [2]. Gingival enlargement is defined by the buildup of extracellular matrix (ECM) within the gingival connective tissue, resulting in an increase in cell and fibrillar element quantity [3]. This phenomenon could be linked to various variables, including an inflammatory stimulation of an undetermined nature. It predominantly impacts anterior teeth rather than posterior teeth and damages vestibular gingival tissue more frequently than lingual or palatal tissue [4].

Gingival enlargement results in cosmetic and functional issues, promotes bacterial biofilm accumulation, and increases susceptibility to periodontal disease. The etiology is multifactorial and may be complex.

Numerous pharmacological agents are documented in the literature as etiological stimuli that cause alterations in the gingival tissue as adverse consequences [5]. Currently, over 20 categories of pharmaceuticals are linked to drug-induced gingival enlargement as an adverse consequence [6].

Phenytoin, cyclosporine, nifedipine, and amlodipine are the medications most linked to drug-induced gingival enlargement [5].

The majority of the evidence is from clinical case reports, with only a limited number of epidemiological studies evaluating the magnitude of this adverse effect [7]. Clinical symptoms of increased volume of the gingival tissues typically emerge within one to three months following the initiation of the related pharmacotherapy. Gingival enlargement typically manifests in the interdental papilla and on the front portion of the labial surfaces. Gingival lobules progressively develop and may exhibit inflammation or fibrosis, contingent upon the extent of inflammation caused by local factors such as oral biofilms [1].

Calcium channel blockers are extensively utilized in the management of hypertension and peripheral vascular disorders [3,4,5,6,7,8]. These medications are linked to specific adverse side effects: vasodilation, peripheral edema, and gastrointestinal problems [8].

Gingival overgrowth is often linked to the usage of nifedipine [2,3,4,5,6,7,8,9]. Amlodipine is a contemporary dihydropyridine calcium antagonist antihypertensive employed in the treatment of hypertension and angina. Amlodipine possesses a prolonged half-life and a diminished side-effect profile compared to nifedipine [6]. Recent investigations indicate that the incidence of gingival enlargement related to amlodipine is considerably lower than that associated with nifedipine [10].

In addition, drug-induced gingival overgrowth (DIGO) is associated not only with morphological changes in the gingival tissues but also with a significant alteration of the oral microbiota. Drugs such as cyclosporine, calcium antagonists, and phenytoin can induce a state of dysbiosis, that is, an imbalance in the composition of the oral microbiota. This state promotes the proliferation of pathogenic bacteria, such as *Porphyromonas gingivalis*, *Fusobacterium nucleatum*, and *Prevotella intermedia*, to the detriment of the physiological microbial species constituting the resident oral flora [11]. Such alteration contributes to the onset of a chronic inflammatory state capable of further aggravating the clinical picture of gingival overgrowth.

To restore oral microbial homeostasis and simultaneously counteract the state of dysbiosis associated with gingival hypertrophy, several innovative natural approaches have been proposed based on the administration of ozone, ozonated water, laser, probiotics, paraprobiotics, and postbiotics [11].

These methods not only aim to rebalance the oral microbiota, but they can also have anti-inflammatory and regenerative effects on the gingival tissues.

This case report aims to describe a case of amlodipine-induced gingival enlargement, emphasizing the significance of a multidisciplinary approach and outlining its therapy across various surgical stages.

## 2. Case Presentation

A 48-year-old non-smoking patient comes to our attention complaining of an abnormal increase in gingival volume. During general anamnestic data collection, the patient reported hypertension, which was successfully managed with calcium antagonists for the past five years, and the daily use of statins to control hypercholesterolemia.

The intraoral objective examination documented local inflammation and gingival overgrowth at the level of the papillae of the upper arch, resulting in partial coverage of the dental crowns (Figure 1A–C).

Additionally, the patient reported spontaneous bleeding from the enlarged gingival tissues. The presence of hypertrophy was not associated with painful symptoms, as expressly reported by the patient.

Addressing the underlying causes of the gingival enlargement, initial steps were provided through professional oral hygiene sessions, which included root planing in the upper quadrants.

At the follow-up visit, 15 days after the professional hygiene treatment, there is evidence of the reduction in the plaque index, the absence of bleeding on probing, but the persistence of the gingival enlargement (Figure 2).

In agreement with the cardiologist, it has been decided to investigate potential interactions between the administration of antihypertensive drugs and the concomitant gingival overgrowth.

The literature extensively documents how nifedipine and other calcium channel blocker drugs used to treat angina, arrhythmias, and hypertension may contribute to the clinical picture of gingival enlargement [12,13,14].

After a careful evaluation together with the referring cardiologist, the drug containing amlodipine has been discontinued, and a new therapy based on olmesartan medoxomil has been initiated.

Fifteen days after the discontinuation of amlodipine, surgical excisions of the interdental hypertrophic gingival tissues were performed, and biopsies were taken and sent to the pathologist for the histological evaluation of the pathologic gingival tissue.

Follow-up visits have been conducted monthly for six months to evaluate the progression of the reduction in gingival overgrowth and the general dento-gingival clinical picture.

The intraoral clinical examination documented the “restitutio ad integrum” of the gingival tissues with full restoration of volume, thickness, and shape of the gingiva with normal marginal festooning and no evidence or signs of inflammation (Figure 3). 

### 2.1. Surgical Procedure

Under local anesthesia with a solution of articaine hydrochloride and adrenaline 1:100,000 (Septanest, Saint-Maur-des-Fossés, Cedex, France), the hypertrophic interdental gingiva located between elements 1.2 and 1.3 and between 1.3 and 1.4 have been excised using a straight micro-blade with a rounded tip (BUSM-6400, Surgistar, Inc., 2310 La Mirada Dr. Vista, CA 92081, USA) (Figure 4A,B).

Once the surgical excision of the hypertrophic gingival tissues has been completed, bleeding has been controlled by positioning a cyanoacrylate-based liquid tissue adhesive (CU-ME-042, Edifici Eureka Campus UAB, 08193, BELLATERRA, Barcelona, Spain) (Figure 5A–C).

Two specimens of gingival tissue, one approximately 4 mm long (Figure 6) and the second 8 mm long (Figure 7), were fixed in saline-buffered formaline and sent to the pathologist to perform the histopathological evaluation and confirm the diagnosis.

After the surgery, post-operative instructions were given to the patient. These included support periodontal therapy (SPT), which foresees domicile oral hygiene and rinsing with 0.2% chlorhexidine digluconate mouthwash twice daily for one week.

After 10 days, a follow-up visit was performed to assess the healing status and check the modifications of the gingival enlargement.

### 2.2. Histopathological Examination

The histopathological analysis confirmed the diagnosis of gingival hypertrophy associated with chorion fibrosis and marked lymphoplasmacytic infiltrate.

Enlarged gingival tissue samples were taken from two different areas: the first between tooth elements 1.2 and 1.3 (Figure 8) and the second between elements 1.3 and 1.4 (Figure 9). Both samples were analyzed to assess the drug-induced tissue changes.

The gingival epithelium shows significant thickening and evident paracheratotic and acanthotic features.

At the level of the chorion, an inflammatory infiltrate of variable density was observed, consisting mainly of lymphocytes and plasma cells.

Areas of fibrosis, characterized by a significant presence of fibroblasts and a significant increase in collagen bundles, were found in the proximity of the inflamed areas (Figure 10A,B).

The presence of cytoid bodies, cytoplasmic structures commonly associated with pathological conditions mediated by inflammatory or degenerative tissue processes, has also been detected (Figure 11).

Indeed, the production of cytoid bodies reflects a cellular response to damage caused by inflammation or immune-mediated tissue destruction.

The histological features found were compatible with a diagnosis of drug-induced gingival hyperplasia.

## 3. Discussion

According to the current literature, gingival overgrowth is an adverse effect commonly associated with drug therapies, particularly calcium channel blockers (CCBs). Seymour et al. state that this condition is characterized by an abnormal increase in the volume of gingival tissue [15,16]. Ellis et al. recognize that the onset of the following condition can also be attributed to the use of other drug classes, such as immunosuppressants and anticonvulsants [7]. As pointed out by Ellis et al., drug-induced gingival overgrowth is a condition that can have significant aesthetic and functional implications, as well as being a potential cofactor in the formation and accumulation of bacterial plaque, primarily involved in the onset and progression of periodontal disease [7,8,9,10,11,12,13,14,15,16,17].

The severity of the condition and the patient’s predisposing factors should guide the treatment of gingival overgrowth [18].

Adverse periodontal effects related to DIGO, including gingival enlargement, occur within three months after the start of drug therapy [19,20] and often regress following discontinuation and replacement of the calcium channel blocker drugs [21].

Drug-induced gingival overgrowth (DIGO) is a condition that can occur at any age, although it is most frequently observed between the ages of 40 and 50 years [22].

Although a correlation between DIGO and sex has not been scientifically confirmed, Gaur and Agnihotri, in a systematic review of the literature, demonstrated that the prevalence of calcium channel blocker-induced gingival overgrowth is higher in men than in women [23]. Many drugs have been associated with the onset of gingival enlargement.

Amlodipine is a third-generation dihydropyridine calcium antagonist that is similar to nifedipine structurally.

For many years, amlodipine was considered a “safe drug” compared to other calcium antagonists, as its use was associated with relatively fewer adverse effects. According to some studies, the incidence of amlodipine-induced gingival overgrowth ranges from 1.7% to 3.3% [22,23,24].

The etiopathogenesis of amlodipine-induced gingival overgrowth is multifactorial. It is based on a combination of molecular and cellular mechanisms and external factors, such as local inflammation induced by bacterial plaque.

Several molecular mechanisms play a key role in the onset of this condition. Amlodipine-induced gingival overgrowth, as demonstrated in this study by histological analysis, is associated with chorion fibrosis and a marked presence of lymphoplasmacytic infiltrate. Also, key histological features include hypertrophy of keratinized epithelium, excessive connective tissue accumulation, and varying degrees of inflammation in the lamina propria [25].

This is consistent with the findings of Lauritano et al. (2019), who demonstrated, in an in vitro study, that gingival fibroblasts exposed to amlodipine overproduce collagen and extracellular matrix proteins, thus contributing to gingival tissue fibrosis [26]. This cellular response appears to be mediated by the activation of molecular pathways, including TGF-β (transforming growth factor-beta), which is known for its role in tissue proliferation and fibrosis. Luciani et al. (2017) demonstrated that the release of pro-inflammatory cytokines, such as IL-6 and TNF-α, increases significantly in the presence of amlodipine, amplifying the cellular activity of gingival fibroblasts [27].

The finding that most patients treated with CCBs do not develop gingival overgrowth led to the discovery of a subset of fibroblasts that are susceptible to CCBs. Genetic predisposition of different fibroblast phenotypes to CCBs may be related to the human lymphocyte antigen [28,29]. However, there is a lack of clinical markers to identify susceptibility to CCBs and patients who are at risk.

The variation in the incidence rate of DIGO among different subjects could depend on an intrinsic genetic predisposition and increased individual susceptibility. Meisel et al. reported that MDR1 (Multi Drug Resistance-1) gene polymorphisms may alter the subject’s inflammatory response to the drug [30].

Other genetic predispositions could influence the metabolism of CCBs, as these drugs are metabolized by the hepatic cytochrome P450 enzymes. Cytochrome P450 genes can show considerable polymorphism, which results in interindividual variation in enzyme activity. This inherited variation in the metabolism of the offending drug may influence the patient’s serum and tissue concentrations [16].

The persistence of the pathological condition, from a molecular point of view, appears to be related to an alteration of cell apoptosis mechanisms [19,20,21,22,23,24,25,26,27,28]. This leads to the persistence and accumulation of overactive fibroblasts, thus favoring the proliferation of over-trophic tissue. As pointed out by Gaur and Agnihotri, the alteration of cell apoptosis mechanisms is probably attributable to the change in intracellular calcium levels, which leads to an increase in the expression of the anti-apoptotic protein Bcl2 [23]. Therefore, the prolonged use of amlodipine, by inducing a reduction in intracellular calcium levels, could play a key role in the onset of this condition.

Among external etiological factors, local inflammation induced by poor control of plaque levels has been recognized as a potential contributory risk factor for the development of gingival overgrowth. Therefore, in order to reduce the incidence of local risk factors, it is essential to educate patients on proper home oral hygiene and motivate them to undergo regular professional oral hygiene sessions, including scaling and root planing.

Supporting this, some case reports showed that oral hygiene control (self-administered and professionally delivered) might reduce DIGO [31]. A cross-sectional study on individuals taking three different CCBs (nifedipine, amlodipine, and felodipine) reported that poor oral hygiene (elevated plaque index) was significantly associated with DIGO [32].

The pharmacological dosage of amlodipine may also have an impact on the onset of DIGO. Therefore, this aspect cannot be ignored, especially in responder patients. As discussed by Gaur and Agnihotri, an amlodipine dosage between 2.5 and 10 mg per day would appear to be associated with gingival tissue thickening [23].

The current scientific literature indicates that the incidence of gingival overgrowth associated with amlodipine is lower than with nifedipine, but the reported cases still suggest the need for active surveillance [33,34,35,36,37].

Seymour et al. pointed out that patients treated with nifedipine presented a more diffuse and severe disease framework than those on amlodipine therapy [15,16]. The findings of Seymour et al. were later confirmed by Bakshi et al. (2023), who recognize that amlodipine has a greater safety profile than first-generation calcium antagonists [16,17,18,19,20,21,22,23,24,25,26,27,28,29,30,31,32,33].

The management of DIGO requires a multidisciplinary approach, including the identification of the etiological trigger of the disease, the adoption of a strict oral hygiene protocol, and, in severe cases, the performance of a biopsy sample for more accurate histopathological investigation [18,19,20,21,22,23,24,25,26,27,28,29,30,31,32,33,34,35,36,37,38].

Before proceeding with biopsy sampling for DIGO, it is advisable to modify or discontinue the drug in question under the supervision of the treating physician, thus eliminating the possible cause [38].

Discontinuation of amlodipine, followed by new drug therapy based on angiotensin II receptor antagonists (ACE inhibitors), proved effective in reducing inflammation associated with gingival overgrowth within approximately 20 weeks [39].

In addition, the integration of probiotics, paraprobiotics, and postbiotics into the daily routine, in the form of mouthwashes or dietary supplements, represents a promising supportive approach to restoring oral microbial homeostasis, preventing drug-induced dysbiosis, a condition potentially capable of aggravating the inflammatory state triggering gingival overgrowth [11].

In conclusion, histopathological analysis confirmed scientific evidence regarding the pathogenetic mechanisms underlying the condition, emphasizing the etiological role of chronic local inflammation and fibrosis of the chorion. An integrated multidisciplinary approach is a successful therapeutical strategy in the management of amlodipine-induced gingival overgrowth.

## 4. Conclusions

Rigorous and proficient surveillance of patients undergoing calcium channel blocker therapy was conducted to promote the early detection of oral complications, including gingival overgrowth. A collaborative approach among the treating physician, dentist, and dental hygienist is essential for the early diagnosis, management, and resolution of gingival enlargement induced by calcium channel blockers through drug substitution, focused oral hygiene sessions, and periodontal surgery.

## Figures and Tables

**Figure 1 diagnostics-15-00320-f001:**
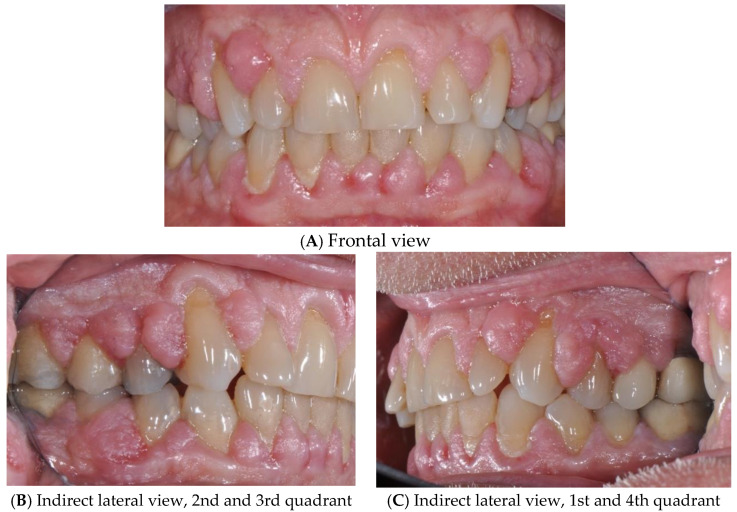
Intraoral clinical examination showing gingival overgrowth: Frontal view (**A**), Indirect lateral view, 2nd and 3rd quadrant (**B**), Indirect lateral view, 1st and 4th quadrant (**C**).

**Figure 2 diagnostics-15-00320-f002:**
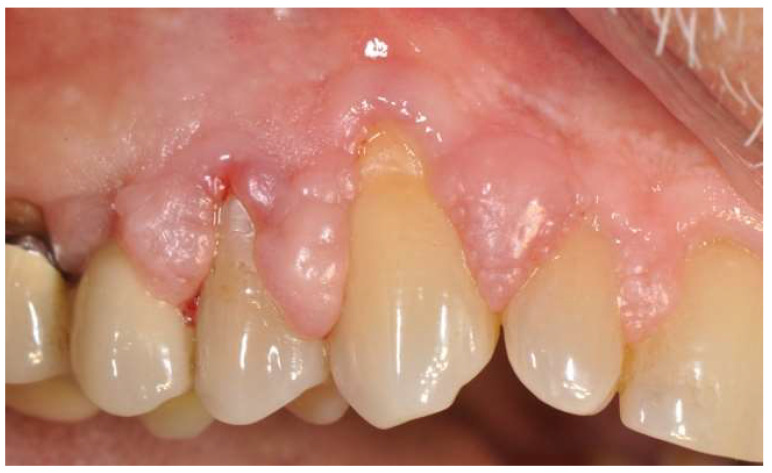
Lateral view, 1st quadrant.

**Figure 3 diagnostics-15-00320-f003:**
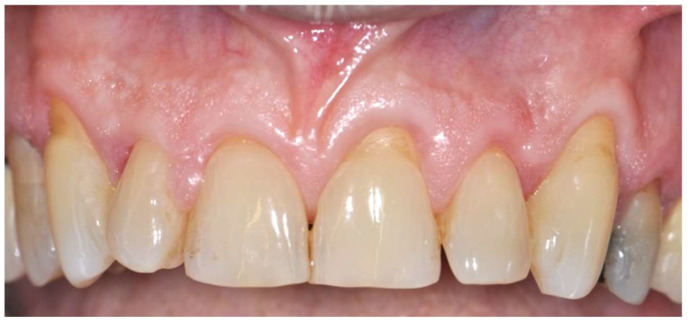
Upper arch in frontal view after 6 months of follow-up.

**Figure 4 diagnostics-15-00320-f004:**
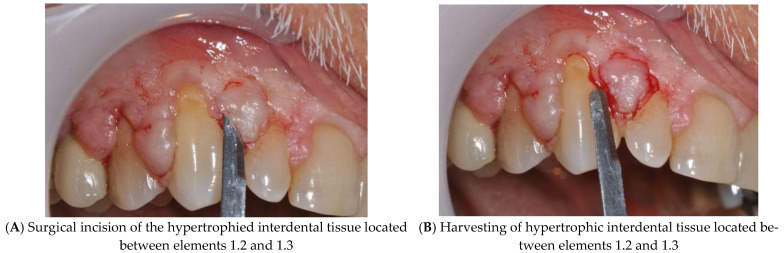
Surgical excision of hypertrophic interdental gingival tissue between elements 1.2 and 1.3: surgical incision (**A**) and tissue removal (**B**).

**Figure 5 diagnostics-15-00320-f005:**
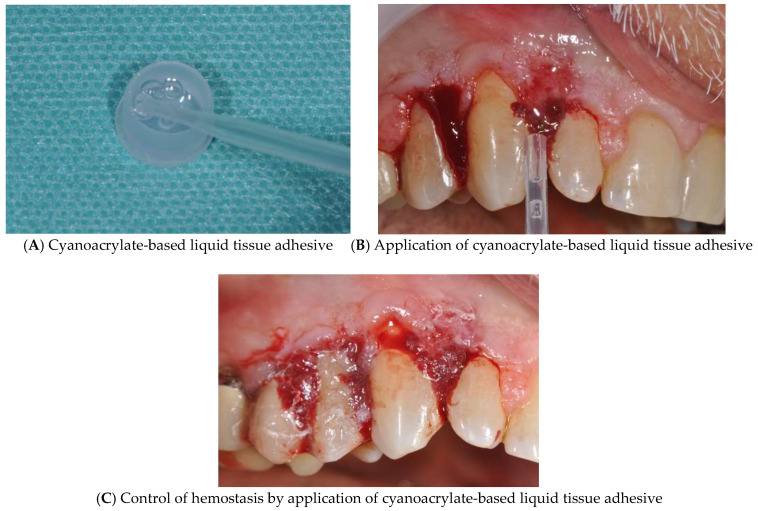
(**A**–**C**) Control of hemostasis a cyanoacrylate-based liquid tissue adhesive.

**Figure 6 diagnostics-15-00320-f006:**
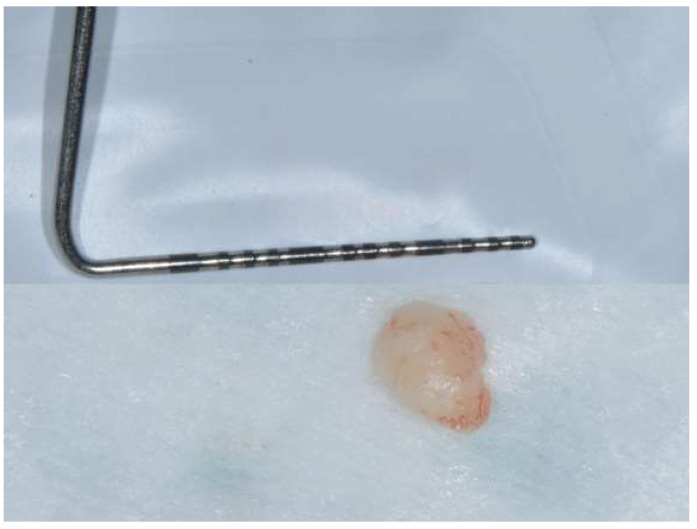
Surgical sample of approximately 4 mm in length.

**Figure 7 diagnostics-15-00320-f007:**
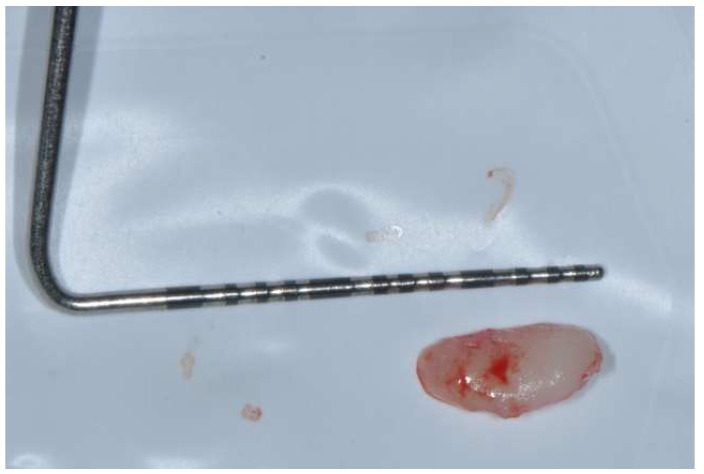
Surgical sample approximately 8 mm in length.

**Figure 8 diagnostics-15-00320-f008:**
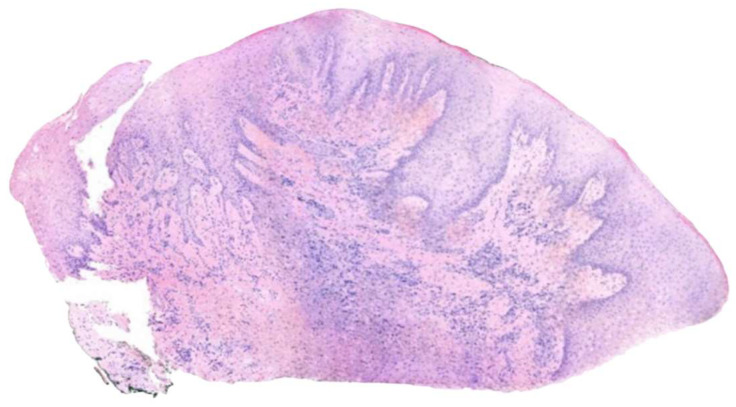
Sample drawn between elements 1.2 and 1.3.

**Figure 9 diagnostics-15-00320-f009:**
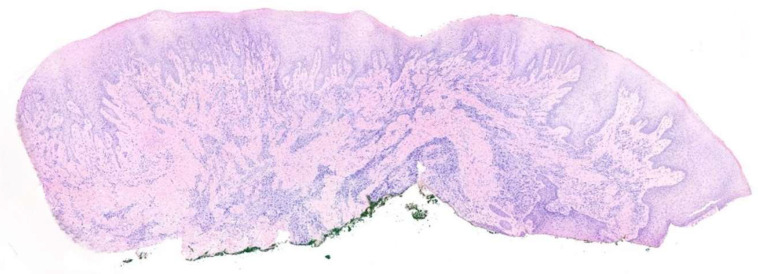
Sample drawn between elements 1.3 and 1.4.

**Figure 10 diagnostics-15-00320-f010:**
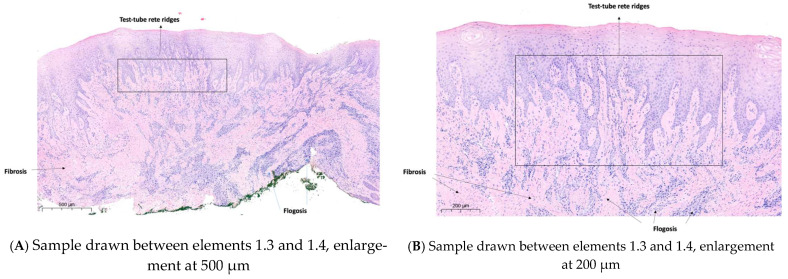
Histological analysis of tissue samples taken between elements 1.3 and 1.4, showing fibrosis, flogosis, and test-tube rete ridges. (**A**) Enlargement at 500 µm; (**B**) Enlargement at 200 µm.

**Figure 11 diagnostics-15-00320-f011:**
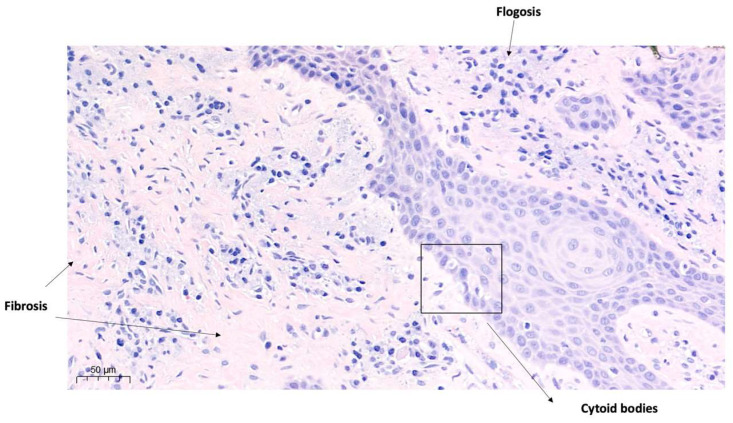
Sample drawn between elements 1.2 and 1.3, enlargement at 50 µm.

## Data Availability

The data presented in this study are available upon request from the corresponding author. The data are not publicly available due to privacy restrictions.

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
