# Peer review of "Calcium Antagonist-Induced Gingival Overgrowth: A Case Report and Literature Review"

_diagnostics, 2025, doi:10.3390/diagnostics15030320_

Round 1

Reviewer 1 Report

Comments and Suggestions for Authors

1.         It is well-known that calcium antagonist could induce gingival overgrowth.

2.         Without compressive reviews in the manuscript, the title revises as “Calcium Antagonist-Induced Gingival Overgrowth: A Case Report”.

3.         Series of clinical images may be put them together as one figure that will be easy to demonstrate the changes of pre- and post-operation.

4.         Series of histopathological images would be put them together as one figure.

5.         The resolution of histopathological images should be improved. 

6.         English editing is required.

7.         Overall, this manuscript is lack of novelty and not well-prepared. 

Comments on the Quality of English Language

English editing is required.

Author Response

Comments 1: It is well-known that calcium antagonist could induce gingival overgrowth.

Response 1: We agree. As expressly stated in the manuscript, DIGO is a well-known topic in the current literature and this case report is clear additional scientific evidence supporting the role of calcium channel blockers in gingival overgrowth or periodontal disease.

Comment 2:  Without compressive reviews in the manuscript, the title revises as “Calcium Antagonist-Induced Gingival Overgrowth: A Case Report”.

Response 2: The original title of the manuscript was “Calcium Antagonist-Induced Gingival Overgrowth: A Case Report”. However, the managing editor suggested changing it to “Calcium Antagonist-Induced Gingival Overgrowth: A Case Report and Literature Review”. While we acknowledge that this is not a comprehensive narrative or systematic review, we have decided to follow the editor’s recommendation and adopt the new title.

Comment 3: Series of clinical images may be put them together as one figure that will be easy to demonstrate the changes of pre- and post-operation.

Response 3: Thank you for pointing this out. Authors agree on the improved clarity achieved by grouping the images. However, according to the authors’ guidelines, images were placed in the manuscript following a chronological order, reflecting the operative protocol applied in the case management.

Comment 4: Series of histopathological images would be put them together as one figure.

Response 4: Separating the histopathological analysis images makes it easier to correlate each image with the description of the histological features.

Comment 5: The resolution of histopathological images should be improved.

Response 5: Images resolution has been improved.

Comment 6: English editing is required.

Comment 6: English editing has been made.

Comment 7: Overall, this manuscript is lack of novelty and not well-prepared.

Response 7: The aim of our manuscript is to support current scientific evidence on this topic and to show the adequate case management.

Reviewer 2 Report

Comments and Suggestions for Authors

Case report of considerable interest for the dental sector through a multidisciplinary approach, requires minor revision before proceeding to publication.

Very well described abstract.

Keywords: sufficient, some are not registered on MeSH, please check and replace them.

Introduction: drug-induced variation of the oral microbiota and all the natural systems that can reduce dysbiosis, such as ozone, ozonated water, laser, probiotics, paraprobiotics, postbiotics (Scribante et al) are missing.

Materials and methods: well detailed and described operative phase with high resolution images including histological examinations.

Discussion: a part of proactive action is missing, which allows to reduce the lesions before the patients started the pharmacological treatment.

Conclusions: modify it based on the changes in the text.

Bibliography; add references requested

Author Response

Comment 1: Very well described abstract.

Response 1: Thank you.

Comment 2: Keywords: sufficient, some are not registered on MeSH, please check and replace them.

Response 2: Thank you for pointing this out. We have changed the manuscript keywors, checking their registration on MeSH terms. Keywords are now: Calcium Channel Blockers, Drug-induced gingival overgrowth (DIGO), Amlodipine, Gingival Overgrowth, Histopathology.

Comment 3: Introduction: drug-induced variation of the oral microbiota and all the natural systems that can reduce dysbiosis, such as ozone, ozonated water, laser, probiotics, paraprobiotics, postbiotics (Scribante et al) are missing.

Response 3: We have adapted the introduction section according to your revision. The suggested reference has been added.

Comment 4: Materials and methods: well detailed and described operative phase with high resolution images including histological examinations.

Response 4: Thank you.

Comment 5: Discussion: a part of proactive action is missing, which allows to reduce the lesions before the patients started the pharmacological treatment.

Response 5: Both the introduction and the discussion section have been improved according to the proposed revisions.

Comment 6: Conclusions: modify it based on the changes in the text.

Response 6: Conclusions have been improved.

Comment 7: Bibliography; add references requested

Response 7: As you suggested, we have added the article by Scribante et al. to the bibliographic list.